Phosphoproteomic insights into the regulation of root length in rice (Oryza sativa L. cv. KDML 105): uncovering key events and pathways involving phosphorylated proteins

Li Rui 1 2
Phaonakrop Narumon 3
Lohmaneeratana Karan 2
http://orcid.org/0000-0003-3696-8390 Roytrakul Sittiruk 3
http://orcid.org/0000-0002-8749-0414 Thamchaipenet Arinthip 4 arinthip.t@ku.ac.th
1 The Graduate School, Kasetsart University , Bangkok , Thailand
2 Department of Genetics, Faculty of Science, Kasetsart University , Bangkok , Thailand
3 Functional Proteomics Technology Laboratory, National Center for Genetic Engineering and Biotechnology, National Science and Technology Development Agency , Pathum Thani , Thailand
4 Omics Center for Agriculture, Bioresources, Food, and Health, Kasetsart University (OmiKU) , Bangkok , Thailand
Fayed Marwa
Electronic publication date: 2025 Jul 4
Publication date: 2025
Volume: 13
Electronic Location ID: e19361
Received 2024 Nov 8; Accepted 2025 Apr 3
Copyright: © 2025 Li et al.
Copyright year: 2025
Copyright holder: Li et al.
License: This is an open access article distributed under the terms of the Creative Commons Attribution License, which permits unrestricted use, distribution, reproduction and adaptation in any medium and for any purpose provided that it is properly attributed. For attribution, the original author(s), title, publication source (PeerJ) and either DOI or URL of the article must be cited.
License URL: https://creativecommons.org/licenses/by/4.0/

Keywords: Maximum root length (MRL), Indoor rice cultivation, Thai jasmine rice, Phosphoprotein profile, Signaling network

Funding: Kasetsart University Research and Development Institute FF(KU)5.64 Interdisciplinary Graduate Program in Bioscience, Faculty of Science, Kasetsart University Bioinformatics Academic Association of Thailand (BAT) Research was funded by grants from Kasetsart University Research and Development Institute [FF(KU)5.64]; the Interdisciplinary Graduate Program in Bioscience, Faculty of Science, Kasetsart University; Bioinformatics Academic Association of Thailand (BAT). The funders had no role in study design, data collection and analysis, decision to publish, or preparation of the manuscript.

==============================
Root is a crucial organ in terrestrial plants, with the maximum root length (MRL) trait of the root system positively correlated with both plant growth and adaptation. However, the mechanisms regulating root length remain inadequately understood due to the dynamics of root growth. Protein phosphorylation precisely regulates various biological processes, providing a pathway to investigate the complex regulatory mechanisms of roots. This study aims to identify key events and pathways that are positively involved in regulating MRL in rice. Using liquid chromatography tandem mass spectrometry (LC-MS/MS), the phosphoprotein profiles of roots exhibiting different MRL phenotypes across three cultivating systems: deep water cultivation (DWC), water agar (WA), and vermiculite-based hydroponics (VBH) were examined. The results showed that the MRL trait of rice is strongly influenced by protein phosphorylation events. Further analysis indicated a clear convergence between phosphorylation signaling and phytohormone signaling in the regulation of MRL. The identified potential phosphoprotein promoters may enhance MRL by promoting root adaptation, optimizing hormonal crosstalk, and facilitating the synthesis of beneficial components. However, given the complexity and dose-dependent nature of hormonal networks, additional quantitative studies were necessary to fully elucidate the mechanisms underlying MRL regulation in rice.

Introduction

Rice (Oryza sativa L.) is a critical global crop that serves as the primary food source for over half of the world’s population. Its relatively small genome of approximately 430 Mb makes it an ideal model organism for studying diverse plant mechanisms. Root is a vital organ in terrestrial plants, playing a key role in anchorage, resource acquisition, and overall plant health. Among various root traits, maximum root length (MRL) is particularly important due to its positive correlation with a plant’s ability to access resources, adapt to environmental conditions, and improve yield performance (Liu et al., 2018; Calleja-Cabrera et al., 2020). Given the global food security challenges, climate change, and soil degradation, understanding the regulatory mechanisms governing rice MRL has become increasingly urgent (Yu et al., 2002; Stuecker, Tigchelaar & Kantar, 2018).

Physiological analysis indicates that the MRL trait in rice undergoes substantial changes during germination (Qin et al., 2022a). Despite the identification of numerous loci across the 12 rice chromosomes (Obara et al., 2010; Wang et al., 2013; Li et al., 2015a), the molecular mechanisms regulating MRL remain inadequately understood. This gap is primarily due to the dynamic nature of root growth in response to complex gene-environment interactions. Environmental factors such as nutrient availability, soil pressure, moisture levels, pH, and temperature significantly influence MRL of rice (Jin et al., 2013; Rativa et al., 2020; Zhang et al., 2020; Du et al., 2021; Fonta et al., 2022). Moreover, the complexity of hormonal regulatory network presents another challenge to elucidating MRL control mechanisms. The primary phytohormones that regulating root development in rice include abscisic acid (ABA), auxin (AUX), cytokinin (CK), ethylene (ETH), and gibberellin (GA). AUX and GA promote root elongation, while ABA, CK, and ETH act as inhibitors. However, the interactions between these hormones are highly complex and context dependent. Their effects can be either synergistic or antagonistic, ensuring the plant finely adjusts root growth to achieve an optimal balance between development and resistance. For instance, the balance between AUX and CK determines whether a plant prioritizes root elongating roots or lateral root development. Under stress conditions, ABA and ETH override the growth-promoting effects of AUX and GA, modulating root growth to conserve resources and enhance stress resistance (Mao et al., 2020; Chen et al., 2022b; Huang et al., 2022; Qin et al., 2022a, 2023). Despite advancements in understanding hormonal and genetic influences, the regulatory mechanisms of MRL remains largely unexplored.

Protein phosphorylation is a reversible post-translational modification (PTM) that rapidly regulates protein activity, localization, and interactions, independently of genomic instructions. This modification often induces functional changes in proteins, contributing to a variety of biological processes, including root elongation (Baskin & Wilson, 1997; Zhang et al., 2013). Protein phosphorylation mediates root growth by regulating hormonal signaling pathways, such as AUX and ETH, without relying on primary receptors (Zhang et al., 2016; D’Alessandro et al., 2019; Han et al., 2021). Additionally, it plays a role in root growth by controlling sucrose phloem transportation (Chen et al., 2022). Hence, studying MRL trait from the perspective of protein phosphorylation provides new insights into the regulatory mechanisms beyond gene expression. By using liquid chromatography tandem mass spectrometry (LC-MS/MS)-based proteomics analysis, various traits of plant roots have been investigated (Marcon et al., 2013; Ghosh & Xu, 2014). However, this approach fails to adequately analyze phosphoprotein profile due to the low abundance of phosphorylated proteins in crude protein extracts. To address this limitation, an enrichment step for phosphoproteins has been introduced (Fíla & Honys, 2012). While phosphoproteomics has been used to explore the regulatory mechanisms of the crown root trait in rice (Singh et al., 2022), the MRL trait remains unstudied.

Indoor cultivation systems, such as deep-water culture (DWC), water agar (WA), and vermiculite-based hydroponics (VBH) are commonly used for studying rice root traits (Trolldenier, 1988; Colmer et al., 1998; Liu et al., 2013). These systems offer a controlled environment by minimizing variations in mineral nutrients, pH, and temperature. However, each cultivation system affects root growth in unique ways. For example, the WA system (0.1% w/v agar) induces an anoxia response (Wu et al., 2017), VBH alters the pressure of the growth matrix (Qin et al., 2022b), and DWC affects stress resistance (Chen, Norton & Price, 2020). Due to significant gene-environment interactions, genetic markers identified for rice MRL are often specific to environmental conditions (Zhang et al., 2001; Wang et al., 2013). This study, therefore, aims to identify key events and pathways linked to the MRL trait in rice under multiple environmental conditions using a phosphoproteomics approach. The results of this study provide a deeper understanding of the regulatory mechanisms governing the MRL trait in rice from a phosphoproteomic perspective and provide valuable insights for optimizing root architecture in crops.

Materials and Methods

Plant materials

Seeds of Thai jasmine rice cultivar Khao Dawk Mali 105 (Oryza sativa L. cv. KDML 105) were obtained from Rice Seed Division, Rice Department, The Ministry of Agriculture and Cooperatives, Thailand. The rice seeds were surface sterilized following modified protocols (Adams & Turner, 2010; Zhou et al., 2020a). Briefly, rice seeds were soaked in sterile deionized water for 30 min, immersed in 70% (v/v) ethanol for 3 min, and then 5% sodium hypochlorite (NaClO) for 10 min, followed by rinsing with sterile deionized water for ten times. Three cultivation systems were prepared in four replicates: (i) deep water culture (DWC; deionized water), (ii) water agar (WA; 0.3% (w/v) agar in deionized water), and (iii) vermiculite-based hydroponics (VBH; vermiculite immersed in deionized water). All cultivation systems were adjusted to pH 6 and placed in sterilized polyethylene containers (17 cm × 10 cm × 7 cm) with a medium depth of 6 cm. Following setup, 100 air-dried seeds were placed into each container. For DWC, seeds were placed on a floating plastic mesh sheet; while for WA and VBH, seeds were directly placed on the surface, as illustrated in Fig. 1. All containers were maintained under germination conditions at 30 °C, with photosynthetically active radiation (PAR) of 200 µmol.m2.s−1 under a 12-h light/dark cycle for 7 days. The experiment employed between-subjects design with each condition tested in four replicates using independent groups.

Figure 1 Framework for preparation of plant materials and total protein samples.

Phenotypic study

The roots of rice grown in three cultivation systems were evaluated for MRL and fresh biomass at the seventh day of germination. These measurements were obtained from 30 seedlings per system. MRL for each biological replicate was measured using a straightedge, while root fresh weight was determined following a surface-drying procedure with paper towels as described by Parmar et al. (2022).

Total protein extraction, phosphoprotein enrichment, and tryptic digestion

Seven days after germination, root samples from 50 rice seedlings per replicate were collected from each of three different cultivation systems, rinsed four times with sterile deionized water, and immediately ground in a sterile mortar with liquid nitrogen. Total protein extraction was performed following an improved TCA/acetone precipitation protocol (Niu et al., 2018). Briefly, 200 mg of fine root powder was vortexed with 0.5% SDS and incubated at room temperature for 30 min. After centrifugation at 12,000 g, 4 °C for 7 min, the supernatant was mixed with 20% TCA/acetone at 1:1 ratio (v/v) and incubated at –20 °C for 1 h prior to centrifugation at 12,000 g, 4 °C for 7 min. The resulting protein pellet was air-dried for 3 min before, resuspending in 100 µl of ultrapure water, and determined the concentration by Lowry method. The quality of protein samples was assessed by 12% SDS-PAGE (Fig. 1). Each protein sample was adjusted to a concentration of 10 mg/ml and then subjected to phosphoprotein enrichment using a Pierce™ Phosphoprotein Enrichment Kit (Thermo Fisher Scientific, Waltham, MA, USA). Reduction of disulfide bonds and alkylation of sulfhydryl groups of the enriched phosphoprotein were performed using 5 mM dithiothreitol and 15 mM iodoacetamide, respectively. The phosphoproteins were then digested with sequencing grade porcine trypsin (Promega, Madison, WI, USA) at 1:20 enzyme-to-substrate ratio and incubated at 37 °C for 16 h.

LC-MS/MS analysis

The tryptic phosphopeptides were injected into an Ultimate3000 Nano/Capillary LC System (Thermo Scientific, Waltham, MA, USA) coupled to a ZenoTOF 7600 mass spectrometer (SCIEX, Framingham, MA, USA). Briefly, 1 μl of phosphopeptides underwent enrichment on a µ-Precolumn (300 µm i.d. × 5 mm) packed with Pepmap 100 resin (5 µm, 100 A, Thermo Scientific, Waltham, MA, USA). Subsequent separation was carried out on a 75 μm I.D. × 15 cm Acclaim PepMap RSLC C18 (2 μm, 100 Å, nanoViper, Thermo Scientific, Waltham, MA, USA). The C18 column was housed in a thermostatted column oven maintained at 60 °C. The chromatographic separation employed solvent A (0.1% formic acid in water) and solvent B (0.1% formic acid in 80% acetonitrile) with a gradient elution of 5–55% solvent B over 30 min, at a constant flow rate of 0.30 μl/min.

On the ZenoTOF 7600 system, source and gas parameters were configured as follows: ion source gas 1 at 8 psi, curtain gas at 35 psi, CAD gas at 7 psi, source temperature at 200 °C with the polarity set to positive and a spray voltage of 3,300 V. Data-dependent acquisition (DDA) was employed, selecting the top 50 most abundant precursor ions per MS1 survey scan for MS/MS, with an intensity threshold of >150 cps. Precursor ions were dynamically excluded for 12 s following two MS/MS acquisitions, with dynamic collision energy enabled. MS2 spectra were collected in the range of 100–1,800 m/z, with a 50-ms accumulation time and the Zeno trap enabled. Collision energy parameters were set to a declustering potential of 80 V, with no DP spread and a CE spread of 0 V. Time bins were summed across all channels using a 150,000 cps Zeno trap threshold. The total cycle time for the Top 60 DDA method was 3 s.

Protein quantitation and identification

Protein quantitation for individual samples was carried out using MaxQuant 2.4.9.0, integrated with the Andromeda search engine (Tyanova, Temu & Cox, 2016a). MS/MS spectra were matched to the UniProt Oryza sativa database. Label-free quantitation was conducted following MaxQuant’s standard settings, including allowance for a maximum of two miss cleavages, a mass tolerance of 0.6 Dalton for main search, and trypsin as the digesting enzyme. Carbamidomethylation of cysteine was specified as a fixed modification, while oxidation of methionine and acetylation of the protein N-terminus were included as variable modifications. Peptide used for protein identification were required to be a minimum of 7 amino acids in length with at least one unique peptide. Proteins were required to be identified by at least two peptides, including one unique peptide. A 1% false discovery rate (FDR) was applied for protein identification, determined using reverse search sequences. The maximal number of modifications per peptide was set to 5. The Oryza sativa proteome downloaded from UniProt on 26 April 2023, served as the reference FASTA file with potential contaminants automatically included to the search space by the software. Log2 transformation of mass intensities was performed using Perseus version 1.6.6.0 (Tyanova et al., 2016b). Missing values were imputed with a constant value of zero. The MS/MS raw data and analysis are available in the ProteomeXchange Consortium via the jPOST partner repository: JPST002995 with dataset identifier PXD050876.

Different expressed phosphoproteins

The visualization and statistical analyses in this study, including heatmap generation, principal component analysis (PCA), and variation analysis, were conducted using MetaboAnalyst version 6.0 (Pang et al., 2022). To improve data quality, features with a relative standard deviation (RSD) greater than 25% were excluded. Data normalization was carried out by applying a base-10 log-transformation. Differentially expressed phosphoproteins (DEPPs) were identified using analysis of variance (ANOVA), with a p-value threshold of 0.1. The normalized peak intensity of identified proteins is visualized by box-and-whisker plots.

Gene Ontology annotation and phosphorylation site prediction

Gene Ontology (GO) annotation was conducted by integrating data from the UniProt knowledgebase (https://www.UniProt.org/proteomes/) (UniProt Consortium, 2022), and Gene3D (http://www.cathdb.info) (Lewis et al., 2018). Phosphorylation site prediction was conducted using the online tool NetPhos version 3.1 (https://services.healthtech.dtu.dk/services/NetPhos-3.1/) (Blom et al., 2004).

Protein-protein interaction

Protein-protein and protein-ligand interactions of DEPPs were investigated using the STITCH version 5.0 (http://stitch.embl.de) (Szklarczyk et al., 2015). A medium confidence level was set with a minimum interaction score of 0.400. Additionally, the analysis was restricted to proteins identified in this study. The network edges represent interaction confidence levels.

Statistical analysis

The statistical analysis of germination data was performed using the R package germinationmetrics version 0.1.8 (Aravind et al., 2019), with three technical replicates, each consisting of 30 seeds. MRL and root fresh weight were analyzed using R version 4.3.2 (R Core Team, 2020). Variance analyses were conducted using one-way ANOVA with Tukey’s honestly significant difference (HSD) post hoc test with 30 biological replicates per treatment (n = 30).

Results

Cultivation systems cause apparent variation in rice maximum root length

After the seventh day of germination, the results showed that the speed of germination was non-significantly accelerated under the water agar (WA) system compared to those of deep water culture (DWC) and vermiculite-based hydroponics (VBH) (Fig. 2A). The total number of germinated seeds did not show a notable difference between the cultivation systems (Fig. 2B). However, both root length (Fig. 2C) and root fresh weight (Fig. 2D) exhibited significant variation across the cultivation systems. Among them, rice seedling grown in the VBH system produced the longest roots and highest fresh underground biomass, followed by the WA system (Fig. 2E).

Figure 2 Physiological response of Oryza sativa L. cv. KDML 105 seeds cultivated under three cultivation systems: deep water culture (DWC), water agar (WA), and vermiculite-based hydroponics (VBH).

(A) germination fitting curve annotated with mean germination time (MGT), (B) number of germinated seeds, (C) maximum root length, (D) root fresh weight, and (E) root phenotype. The scale bar represents 5 cm.

Cultivation systems induce significant differences in the phosphoprotein profiles of roots

To examine the global phosphorylation dynamics associated with the MRL trait, total proteins were isolated from roots grown in the DWC, WA, and VBH cultivation systems. Label-free quantification using LC-MS/MS identified a total of 18,423 phosphoproteins from roots across the three systems (Table S1). After data normalization, 13,255 proteins (Table S2) were submitted for further analysis. Significant differences in phosphoprotein profiles among DWC, WA, and VBH systems were revealed through heatmap analysis (Fig. S1). Partial least squares-discriminant analysis (PLS-DA) classified the phosphoprotein profiles into three distinct groups along the directory of component 1 (Fig. 3).

Figure 3 The 2D score plot of partial least squares-discriminant analysis (PLS-DA) displays phosphoprotein data from rice roots grown under different cultivation systems:.

deep water culture (DWC), water agar (WA), and vermiculite-based hydroponics (VBH). Each dot represents the phosphoprotein profile of a sample.

Analysis of differentially expressed phosphoproteins and Gene Ontology

In this study, roots obtained from both WA and VBH systems exhibited a significantly longer MRL compared to that of DWC. To identify DEPPs, a comparative analysis was conducted using ANOVA with a p-value threshold of 0.1. The results revealed that five out of the 13,255 identified phosphoproteins exhibited significant changes in abundance across the different cultivation systems (Fig. 4). The peak intensities of Q8L4D3 (heat shock protein 40/DnaJ), Q2R0A3 (leucine-rich repeat family protein/LRR), and Q5SN55 (DUF6598 domain-containing protein) were significantly elevated in roots from both WA and VBH systems. In contrast, the peak intensity of Q0D6I4 (RNA polymerase-associated protein C-terminal repeat protein 9/Crt9) and Q8H5D5 (uncharacterized protein) were notably higher in roots from VBH system (Fig. 5, Table S3).

Figure 4 One-way ANOVA plot illustrates the comparison of identified phosphoproteins from rice roots cultivated in water agar (WA) and vermiculite-based hydroponics (VBH) vs. those of deep-water culture (DWC).

Q8L4D3, heat shock protein 40/DnaJ; Q2R0A3, leucine-rich repeat family protein/LRR; Q5SN55, DUF6598 domain-containing protein; Q0D6I4, RNA polymerase-associated protein C-terminal repeat protein 9/Crt9; and Q8H5D5, uncharacterized protein. Colored dots, phosphoproteins with statistically significant different expression (p-value < 0.1); gray dots, phosphoproteins that do not meet the significant threshold.

Figure 5 Boxplots illustrate the summary of normalized peak intensity for differentially expressed phosphoproteins (DEPPs) detected in rice roots grown in deep-water culture (DWC), water agar (WA), and vermiculite-based hydroponics (VBH) cultivation systems.

(A) Q0D6I4, RNA polymerase-associated protein C-terminal repeat protein 9/Crt9, (B) Q8H5D5, uncharacterized protein, (C) Q8L4D3, heat shock protein 40/DnaJ, (D) Q2R0A3, leucine rich repeat family protein/LRR, and (E) Q5SN55, DUF6598 domain-containing protein. Each box represents the interquartile range (IQR), with whiskers extending 1.5 times within the IQR. Black dots represent individual data points, while yellow notch indicates a confidence interval around the median. The comparison is made with Fisher’s LSD with a p-value at 0.05.

GO investigation (Table 1) showed that Q0D6I4, Q8L4D3, and Q2R0A3 were associated with the paf1 complex (GO:0016593), the cytosol (GO:0005829), and the plasma membrane (GO:0005886). These proteins are involved in transcription elongation by RNA polymerase II (GO:0006368), protein metabolism (GO:0019538), and the plant-type hypersensitive response (GO:0009626), respectively. However, the other two DEPPs have yet to be annotated. All five proteins were predicted to contain phosphorylation sites using NetPhos.

Table 1 Gene ontology (GO) annotation of the five significantly changed phosphorylated proteins associated with the maximum root length (MRL) trait in rice using UniProt and Gene3D databases.

ID	Protein name	Biological process	Cellular component	Molecular function	
Q8H5D5	Uncharacterized protein	None	None	None	
Q0D6I4	RNA polymerase-associated protein C-terminal repeat protein 9	Transcription elongation
by RNA polymerase II (GO:0006368)	Cdc73/Paf1 complex (GO:0016593)	RNA polymerase II complex binding (GO:0000993)	
Q5SN55	DUF6598 domain-containing protein	Regulatory processes	None	None	
Q8L4D3	Heat shock protein 40/DnaJ	Protein folding (GO:0006457), Stress response (GO:0006950)
and Protein metabolic
process (GO:0019538)	Cytosol (GO:0005829)	Protein binding (GO:0005515)	
Q2R0A3	Leucine rich repeat family protein	Plant-type hypersensitive response (GO:0009626)
and Signal transduction (GO:0007165)	Plasma membrane (GO:0005886)	Protein binding (GO:0005515)	

Protein-protein interaction analysis of DEPPs

To explore the critical biological processes involved with the potential MRL regulatory phosphoproteins identified in this study, a protein-protein interaction analysis was performed using STITCH. While no information was retrieved for Q5SN55 and Q8H5D5 from the Oryza sativa japonica protein database in STITCH (Table S4), three phosphoproteins Q0D6I4 (RNA polymerase-associated protein C-terminal repeat protein 9/Crt9), Q8L4D3 (heat shock protein 40/DnaJ), and Q2R0A3 (leucine rich repeat family protein/LRR) were found to be part of a network involving key root growth hormones, including abscisic acid (ABA), auxin (AUX), cytokinin (CK), ethylene (ETH), and gibberellin (GA). This interaction network also encompassed several biochemical components, such as adenosine triphosphate (MgATP), adenosine diphosphate (MgADP), cyclic guanosine monophosphate (cGMP), and cyclic adenosine monophosphate (cAMP), as well as enzyme like nitrilase (NIT) and aldehyde dehydrogenase (ALDH), as illustrated in Fig. 6. This network highlights the potential integration of hormonal signaling and metabolic pathways in regulating MRL in rice.

Figure 6 The protein-protein interaction network associated with maximum root length (MRL) trait.

Three of the five significantly differentially expressed phosphoproteins Q0D6I4 (RNA polymerase-associated protein C-terminal repeat protein 9/Crt9), Q2R0A3 (leucine-rich repeat family protein/LRR), and Q8L4D3 (heat shock protein 40/DnaJ) integrating with predicted network patterners including key phytohormones (abscisic acid, auxin, cytokinin, ethylene, and gibberellin), adenosine diphosphate (MgADP), adenosine triphosphate (MgATP), cyclic adenosine monophosphate (cAMP), cyclic guanosine monophosphate (cyclic GMP), aldehyde dehydrogenase (ALDH), and nitrilase (NIT). Protein names are labeled in black. Identified phosphoproteins in this study are highlighted with red rectangle. Line thickness corresponds to the strength of the interactions. The confidence level of the interactions for high, moderate, and low (reported interactions but lack strong supporting evidence) are shown in red, green, and gray lines, respectively.

Discussion

The MRL trait is highly responsive to phosphorylation events

In this study, although no significant effects on germination were observed, the rice root length trait exhibited notable differences across cultivation systems. This highlights the sensitivity of rice maximum root length (MRL) trait to environmental variation. Specifically, vermiculite-based hydroponics (VBH) proved to be the most beneficial condition for the MRL trait, followed by water agar (WA), while deep water culture (DWC) being the least advantageous. Previous analyses have identified distinct environmental conditions for each system: the DWC system increases humidity (Zhang et al., 2001), the WA system reduces oxygen availability (Wu et al., 2017), and the VBH system alters the pressure of the growth matrix (Qin et al., 2022b). Despite the absence of a thorough comparative analysis of the physical attributes of these cultivation systems, the VBH system stands out due to its significant light blockage, resulting in a considerably darker growth environment. It has been demonstrated that darkness enhances MRL in rice by modulating the auxin (AUX) signaling pathway, a process regulated by protein phosphorylation (Wang, Ho & Chen, 2011; Sassi et al., 2012; Ki, Sasayama & Cho, 2016). These findings highlight the role of protein phosphorylation in regulating root growth and may provide an explanation for the enhanced MRL observed in the VBH system. The considerable variation in MRL phenotype across different cultivation systems underscores the importance of considering the influence of cultivation conditions in future studies on rice root structure and development.

Furthermore, PCA analysis categorizes the phosphoprotein profiles of roots from different systems into three distinct groups, thereby revealing the differences in protein phosphorylation patterns among these systems (Fig. 3). These differences suggest that variations in factors such as light, humidity, and oxygen content within cultivation systems may significantly influence the phosphoprotein profiles of rice roots. However, these variations in the systems induced only minimal changes in the abundance of phosphoproteins, with just five showing significant alterations (Fig. 4). This underscores the strong responsiveness of the rice MRL trait to fluctuations in phosphorylation events, aligning with previous phosphoproteomic studies on root traits in Arabidopsis, which have demonstrated that even minor phosphorylation changes can significantly alter root traits (Zhang et al., 2013; D’Alessandro et al., 2019). This sensitivity indicates the potential to modulate MRL traits in rice through precise control of phosphorylation pathways.

Phosphorylated leucine-rich repeat protein positively regulates the MRL trait

Plant disease resistance (R) family proteins play a crucial role in both plant defense and environmental adaptation. In particular, nucleotide-binding leucine-rich repeat (NB-LRR) type R proteins possess an LRR domain that detects extracellular biotic and abiotic signals, while the NB domain triggers immune responses, including programmed cell death (PCD), after binding to ATP or GTP (Rairdan et al., 2008; Yang et al., 2022; Hussain et al., 2024). To prevent the activation of unnecessary defense responses, NB-LRR proteins are regulated by a conserved nucleotide-binding APAF-1, various R-proteins and CED-4 (NB-ARC) domain (Okuyama et al., 2011). Disturbance of the auto-inhibition function of the NB-ARC domain within a coiled-coil (CC)-NB-LRR protein impeded root elongation in rice (Yu et al., 2018). In the present study, a CC-NB-LRR protein Q2R0A3 was found to be highly abundant in the roots grown in both WA and VBH systems (Fig. 5), which exhibited longer MRL. This suggests that the observed phosphorylation event has no impact to the auto-inhibition of the LRR protein. The identified CC-NB-LRR protein appears to influence the MRL trait through a different mechanism rather than triggering the innate immunity responses. However, further investigations of the protein functions and their associations with root growth should be conducted to evidently prove that protein phosphorylation plays an important role in the MRL trait of rice plants.

The phytohormone ABA plays a significant role in the environmental adaptation of plants. Early investigations have shown that the expression of NB-LRR proteins responds to environmental stimuli (Ariga et al., 2017). Artificially enhancing NB-LRR protein expression elevated both the synthesis and the sensitivity of ABA (Xun et al., 2019; Li et al., 2024). Based on our results, the phosphorylated CC-NB-LRR protein may promote rice MRL by improving the roots’ ability to sense environmental factors together with the phytohormone ABA acting as a signaling molecule to initiate downstream adaptation responses. This includes the biosynthesis of glycine betaine and AUX, driven by the predicted network partners, aldehyde dehydrogenase (ALDH) and nitrilase (NIT) (Fig. 6) (Ishitani et al., 1995; Rahman, Miyake & Takeoka, 2002; Niu et al., 2014; Man, 2016; Lehmann et al., 2017).

Phosphorylated heat shock protein benefits the MRL trait by optimizing hormonal crosstalk

Phytohormones are small organic compounds that regulate physiological responses and gene expression in a dose-dependent manner (Li et al., 2015b). Their complex interactions and integrated signaling networks enable effective coordination of plant growth and adaptation. ABA, AUX, CK, ETH, and GA are key phytohormones that influence the rice MRL trait. Individually, exogenous ABA promotes root length by inducing expansion in the root tip cells (Chen et al., 2006). AUX increases the root elongation zone (Qi et al., 2012), while GA supports both root cell elongation and division (Sauter & Kende, 1992). Conversely, CK inhibits root length by reducing the size of meristem (Wang et al., 2020), and ETH decreases cell wall plasticity (Zhou et al., 2024). However, the roles of these phytohormones in regulating the rice MRL trait are governed by a complex network. Although both ABA and AUX generally promote root elongation, high amount of ABA also enhance AUX synthesis and accumulation, which eventually inhibits rice MRL (Yin et al., 2011; Qin et al., 2023). Additionally, the inhibitory effect of ETH on the MRL requires cooperation with ABA and AUX (Ma et al., 2014; Huang et al., 2022), while CK primarily exerts its inhibitory effect by increasing ETH levels (Zou et al., 2018). In this intricate network, GA plays a significant role in counteracting the inhibitory effects of ethylene and CK on the rice MRL trait (Zhou et al., 2020b; Qin et al., 2022a).

In this study, the abundance of phosphorylated Q8L4D3 is positively associated with the MRL phenotype in rice, particularly in the roots from WA and VBH cultivation systems (Fig. 5). Q8L4D3 is a DnaJ protein (heat shock protein 40, Hsp40) characterized by its J-domain, which interacts with DnaK (Hsp70) chaperones to regulate protein folding by stimulating the ATPase activity of DnaK (Walsh et al., 2004). The expression levels of DnaJ and DnaK respond to the environmental stimuli (Zhou et al., 2023). Enhancing the expression of DnaJ proteins markedly increased MRL in Arabidopsis and Medicago sativa L. under stress conditions (Zhichang et al., 2010; Liu et al., 2023). In rice, both morphological development and environmental adaptation are positively associated with DnaJ expression, likely due to its regulatory role in the crosstalk between AUX, CK, and GA (Wang et al., 2019, 2022). Similar results have been observed in Arabidopsis and yeast (Fliss et al., 1999; Bekh-Ochir et al., 2013), highlighting the conserved role of DnaJ in growth regulation through mediate hormonal signaling. From protein-protein interaction analysis (Fig. 6), nitrilase (NIT) and cGMP were identified as key partners that influence root growth by modulating AUX signaling (Nan et al., 2014; Lehmann et al., 2017). Furthermore, the activities of DnaJ and DnaK are regulated by phosphorylation (Kostenko, Jensen & Moens, 2014; Zheng et al., 2016). Hence, Q8L4D3 may enhance rice MRL by optimizing the balance of AUX, CK, and GA signaling pathways (Fig. 7).

Figure 7 Summary of critical signalings and metabolic pathways involved in phosphorylated proteins responding to rice MRL trait in water agar (WA) and vermiculite-based hydroponics (VBH) cultivation systems.

Phosphoproteins, predicted proteins, hormones, and chemical regulators are indicated in red, plum, light blue, and purple, respectively. Crt9, RNA polymerase-associated protein C-terminal repeat protein 9; DnaJ, heat shock protein 40; LRR, leucine-rich repeat family protein; ABA, abscisic acid; GA, gibberellin; ALDH, aldehyde dehydrogenase; NIT, nitrilase; H3K4me3, H3 lysine 4 trimethylation. Yellow arrows indicate the interaction of DnaJ with phytohormone ABA, AUX, and GA, while yellow dotted arrows indicate the crosstalk among these phytohormones.

Phosphorylated RNA polymerase-associated protein Ctr9 promotes MRL trait

The polymerase-associated factor 1 complex (Paf1C) is composed of Cdc73, Ctr9, Leo1, Paf1, and Rtf1 (Chen et al., 2022a). This complex plays a vital role in regulating RNA polymerase II (Pol II)-mediated transcription by influencing histone modifications such as methylation, ubiquitination, and phosphorylation, thereby affecting chromatin structure and gene expression (Wier et al., 2013; Strikoudis et al., 2017; Ropa et al., 2018; Hou et al., 2019). In this study, the Paf1C component Ctr9 (Q0D6I4) was abundantly identified, particularly in root growth under VBH cultivation system (Fig. 5). Ctr9 is essential for morphological development (Shiraya et al., 2008; Ouna et al., 2012), potentially due to its regulatory role in the H3 lysine 4 trimethylation (H3K4me3) (Chaturvedi et al., 2016; Oh & Lee, 2016). A positive correlation between Ctr9 and H3K4me3 has been established (Bahrampour & Thor, 2016). The effects of H3K4me3 on plant growth and adaptation have been extensively studied by Foroozani, Vandal & Smith (2021). In Arabidopsis, H3K4me3 showed a positive correlation with MRL trait (Yao et al., 2013). Although phosphorylation has been shown to significantly affect the H3K4me3 histone modification (Andrews et al., 2016; Harris et al., 2023), a direct link between phosphorylated Ctr9 and H3K4me3 remains to be established.

Additionally, the protein network enrichment analysis of this study revealed a direct association between cAMP, cGMP, and Ctr9 (Fig. 6). Previous studies demonstrated that both cAMP and cGMP facilitate root growth by influencing hormonal signaling pathways, a process supported by phosphorylation (Isner, Nühse & Maathuis, 2012; Domingo et al., 2024). Specifically, cAMP promotes root growth via an ABA-dependent mechanism that upregulates defense proteins involved in the adaptive response, including those related to GA synthesis (Uematsu et al., 2007; Zhao et al., 2021; Wang et al., 2024). Conversely, cGMP modulates root growth by affecting AUX signaling through cGMP-dependent protein kinase (PKG) (Nan et al., 2014). Furthermore, the metabolism of phenylpropanoid-chemical components that enhances root elongation-is regulated by both cAMP and cGMP (Pietrowska-Borek & Nuc, 2013; Thwe et al., 2016). Based on these early findings, it can be hypothesized that the positive correlation between phosphorylated Ctr9 and the rice MRL trait may result from modulation of H3K4me3, hormone signaling (AUX and GA), and phenylpropanoid, as illustrated in Fig. 7. However, further investigation into the relationships among phosphorylated Ctr9, cAMP, and cGMP is needed.

Conclusions

This study demonstrates that the MRL trait in rice is tightly regulated by protein phosphorylation events. Alterations in the abundance of a few phosphoproteins can lead to significant changes in the root phenotype of rice. Five phosphoproteins consistently associated with MRL were isolated across DWC, WA, and VBH systems. Our analysis results revealed a strong connection between phosphorylation signaling and hormone signaling in the regulation of the MRL. Phosphoprotein related to AUX and GA biosynthesis, phenylpropanoid production, and histone H3 lysine 4 trimethylation may positively correspond with rice MRL. Further functional validation of these upregulated phosphoproteins will be required to prove their roles in regulation of the MRL trait. These phosphoproteins have the potential to serve as biomarkers for further research into root length traits, providing new avenues for enhancing rice resilience and productivity through targeted genetic or environmental modifications that modulate phosphorylation signaling pathways.

Supplemental Information

Supplemental Information 1 Heatmap illustrates the intensity of all 13,255 identified phosphoproteins from rice roots cultivated in deep water culture (DWC), water agar (WA), and vermiculite-based hydroponics (VBH).

Supplemental Information 2 The raw numerical data were used in statistical analysis to identify significantly changed phosphoproteins among rice roots harvested from different cultivation systems.

Deep water culture (DWC), water agar (WA), and vermiculite-based hydroponics (VBH).

Supplemental Information 3 UniProt IDs of identified phosphoproteins after data preprocessing.

Supplemental Information 4 Significantly different expression of 5 phosphoproteins in rice roots grown in DWC, WA, and VBH cultivation systems determined by ANOVA with a p-value at 0.1.

Deep-water culture (DWC), water agar (WA), and vermiculite-based hydroponic (VBH).

Supplemental Information 5 Conversion of protein IDs from UniProt into Gene IDs used by the small molecules and protein interaction STITCH database.

Supplemental Information 6 The maximum root length (in cm) of germinated rice seedlings.

Supplemental Information 7 The root fresh weight (in mg) of germinated rice seedlings.

Supplemental Information 8 The number of germinating seeds.

Supplemental Information 9 Germination speed and percentage.

We are very grateful to the Rice Seed Division, Rice Department, Ministry of Agriculture and Cooperatives, Thailand for providing seeds of the Thai jasmine rice cultivar Khao Dawk Mali 105 (Oryza sativa L. cv. KDML 105).

Additional Information and Declarations

Competing Interests

The authors declare that they have no competing interests.

Author Contributions

Rui Li conceived and designed the experiments, performed the experiments, analyzed the data, prepared figures and/or tables, authored or reviewed drafts of the article, approved the final draft, and approved the final draft.

Narumon Phaonakrop performed the experiments, prepared figures and/or tables, approved the final draft, and approved the final draft.

Karan Lohmaneeratana performed the experiments, prepared figures and/or tables, approved the final draft, and approved the final draft.

Sittiruk Roytrakul conceived and designed the experiments, analyzed the data, authored or reviewed drafts of the article, approved the final draft, and approved the final draft.

Arinthip Thamchaipenet conceived and designed the experiments, analyzed the data, authored or reviewed drafts of the article, approved the final draft, and approved the final draft.

DNA Deposition

The following information was supplied regarding the deposition of DNA sequences:

The MS/MS raw data are available in the ProteomeXchange Consortium via the jPOST partner repository: JPST002995, PXD050876.

Data Availability

The following information was supplied regarding data availability:

The raw numerical data are available in the Supplemental Files.

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
