# Peer review of "Phosphoproteomic insights into the regulation of root length in rice (Oryza sativa L. cv. KDML 105): uncovering key events and pathways involving phosphorylated proteins"

_PeerJ, doi:10.7717/peerj.19361_

## Round 0.1 · original submission · Minor Revisions

· Academic Editor

Minor Revisions

Kindly revise your manuscript carefully.

Thanks

·

Basic reporting

1-Your study on the regulatory role of phosphorylation mechanisms in controlling maximum root length (MRL) presents a significant and innovative contribution. Highlighting the effects of phosphorylation on hormonal signaling and environmental adaptation processes further enhances the agricultural relevance of your work. However, clarifying the research hypothesis could make the focus and key takeaways of your manuscript more explicit.

2-The research hypothesis should clearly specify the mechanisms that phosphorylation is proposed to influence. For example, in the introduction, a more specific hypothesis regarding the effects of phosphorylation on root length could be articulated. In the discussion section, the contributions of phosphorylation to direct growth effects and adaptation processes should be more distinctly separated.

3-While the data are well-organized overall, the protein-protein interaction network in Figure 6 appears overly dense and challenging to interpret. A simplified visualization would make it easier for readers to identify the critical interactions within the network. Additionally, providing more descriptive captions for all figures and tables could strengthen the visual presentation of the manuscript.

Experimental design

1-The physiological differences between cultivation systems and their effects on phosphoproteomic profiles do not appear to be clearly explained. Specifically, factors unique to cultivation conditions, such as oxygen levels, ion concentrations, or other physicochemical parameters, could be discussed in greater detail regarding their influence on phosphoprotein expression.

Validity of the findings

1-The chosen p-value threshold (p < 0.1) is relatively lenient for biological studies and may increase the likelihood of false positives. Adopting a stricter threshold (p < 0.05) would likely enhance the reliability and statistical significance of the findings.

Additional comments

1. Subheading Adjustment:
The subheading "Phenotype study" could be replaced with "Phenotypic analysis" as it better reflects the academic tone and accurately describes the content.

2. Line 222:
The phrase "did not showed" contains incorrect verb usage. The correct form is "did not show". Fixing such errors will enhance the professional fluency of the manuscript.

3. Terminology in Line 190:
Instead of "logarithmic transformation," the more commonly used term "log-transformation" should be preferred. This adjustment aligns better with standard academic terminology.

4. In the Abstract:
The expression "positively correlating" appears multiple times. It is recommended to use "positively correlated with" instead, as it is a more precise and widely accepted phrasing in academic writing.

5. Subject-Verb Agreement in Line 247:
In this sentence, the subject is plural; therefore, "was" should be replaced with "were significantly elevated." Correcting such errors ensures the grammatical accuracy and quality of the manuscript.

Reviewer 2 ·

Basic reporting

it has good language and clear idea. The structure also good

Experimental design

In general well designed but I have some points that need to be clarified

Material and methods
What is the experimental unit please clarify? How many seeds or plant per unit? Per replicate?

Used 100 seeds for what ? It is not clear, is it per system? for all three systems?

The three different system types have different density which impacts on root development and signal transfer between the root cells and the microenvironment. This could impact the results when comparing these three systems. Doing this kind of experiment should be on same system and change only one parameter.

Why did not use normal soil as a control ? and to simulate the real condition

Why did not do RNAseq to determine the up and down regulated genes directly from the mRNA?

Total protein extraction is not clear whether it was based on one individual plant per replicate or not?
In line 130 extract 200 mg root……, in line 136 of protein subjected ….. could researcher explain how 200mg of root give 200 mg protein ?

Validity of the findings

The finding will help the rice researcher community to develop rice varieties that can adapt to climate change and drought environment with limited water resources.
Also will help understanding the genes expression that control root length in response to different stress and identify the metabolic pathway

Data and statistic analysis sounds good and sufficient.

Conclusion well written and summarized the finding

Reviewer 3 ·

Basic reporting

The manuscript is well written in the acceptable format and relevant references are cited. The English is good in general. The figures are clear. The raw data on phosphorylated proteins are submitted. However, the authors need more data to prove their hypothesis presented in the manuscript.

Experimental design

The authors comparatively analyzed the protein phosphorylation profiles in rice roots harvested from the three culture systems [deep water culture (DWC), water agar (WA), and vermiculite-based hydroponics (VBH)]. As the authors concerned (L85-88), each culture system may have various distinct effects on root growth. Some of the different expressed phosphoproteins (DEPPs) detected might not be related to the maximum root length (MRL) trait in rice plants. To find DEPPs associated with root growth, the authors may need to collect data from more types of culture systems, as mentioned in the conclusion part (L389-391). In addition, comparative analysis of two or more rice cultivars may be also beneficial.

Validity of the findings

The authors discussed about possible associations of CC-NB-LRR protein Q2R0A3 and DnaJ protein Q8L4D3 with the MRL trait. However, to do such discussions, investigations of the protein functions and its associations with root growth should be conducted. At least from the current data, it is difficult to discuss whether and/or how phosphorylation of the proteins is important in regulation of root growth.
Taken together, the authors need additional data and/or strong evidence to prove their hypothesis that protein phosphorylation plays an important role in the MRL trait of rice plants.

Additional comments

I have no additional comment.

Reviewer 4 ·

Basic reporting

Introduction:
Line 47: “Positively correlation” Should be "positive correlation."
Line 59-61: The phytohormones namely abscisic acid ……. development in rice." Consider reframing this sentence “The primary phytohormones regulating root development in rice include abscisic acid (ABA), auxin (AUX), cytokinin (CK), ethylene (ETH), and gibberellin (GA)."
Line 63: "synergistic and antagonistic" should be "synergistic or antagonistic".
Use active voice where possible for clarity.
Break long sentences to enhance readability.
Reduce redundancy, especially in discussing hormonal regulation and cultivation systems.
The introduction does not clearly establish the research gap. It would be better to add a sentence after discussing hormonal regulation to highlight that "despite advancements in understanding hormonal and genetic influences, phosphoproteomic regulation of MRL remains largely unexplored."

Experimental design

Line 102-105: "Seeds of Thai jasmine rice ……. Cooperatives, Thailand)..." Parentheses should not enclose excessively long clauses. Rephrase as… "Seeds of Thai jasmine rice cultivar Khao Dawk Mali 105 (Oryza sativa L. cv. KDML 105) were obtained from the Rice Seed Division, Rice Department, Ministry of Agriculture and Cooperatives, Thailand."
Line 122: "MRL for each biological replication was measured using a straightedge..." “replication” replace with “replicate”.

Validity of the findings

The Results section should focus on describing the findings objectively. Some interpretative statements should be moved to the Discussion section. Such as… “These five upregulated phosphoproteins are assumed to be positively associated with MRL”. This is an interpretation and could fit better in the Discussion.
Line 244: "Five out of the 13,255 identified phosphoproteins …. changes in abundance." This number seems surprisingly low. Were only highly stringent thresholds used? A brief clarification of the selection criteria for DEPPs would improve transparency.
The assumption that certain phosphoproteins directly regulate MRL should be made carefully. For example, "These five upregulated phosphoproteins are assumed to be positively associated with MRL (line 250)," instead used a more rigorous phrasing: "These five upregulated phosphoproteins are correlated with increased MRL and may play a role in its regulation, pending further functional validation."

Additional comments

Discussion:
The discussion presents important findings but is sometimes dense with technical details, making it hard to follow. Simplifying certain phrases and improving transitions between ideas would enhance readability.
Some references to figures are unclear. Such as, "Phosphoproteomics analysis showed consistent results, …. different systems (Fig. 3)" consistent in what way? It would be better to briefly summarize the key findings from the figure.
Some claims need more obvious justification. For example, the conclusion that phosphorylation of Ctr9 correlates with the MRL trait should be stated with applicable limitations (correlation does not imply relationship).
Before discussing the role of phosphoproteins, introduce why phosphorylation is critical for root development.
When moving from one protein to another, add a brief transition linking their relevance to MRL rather than abruptly introducing a new topic.
Suggestions:
Proofread for grammar and clarity.
Improve logical flow by structuring sections with clearer transitions.
Some sentences are overly long and complex. Breaking them down would make the arguments clearer.
Some verb tenses are inconsistent, such as shifting between past and present tense within the same paragraph.

Reviewer 5 ·

Basic reporting

No comment.

Experimental design

No comment.

Validity of the findings

The author need to restructuring this conclusion section due to some sentences have referral to Figure.

Additional comments

The manuscripts have several grammatical error. Several references were too old which more than 10 years.

Annotated reviews are not available for download in order to protect the identity of reviewers who chose to remain anonymous.

---

## Round 0.2 · accepted · Accept

· Academic Editor

Accept

Thanks for following all the comments previously sent by the reviewers. Now, your manuscript is suitable for publication.

·

Basic reporting

The manuscript is clearly written and well-structured. The authors have addressed all previous concerns regarding clarity, data presentation, and adherence to reporting standards. Figures and tables are appropriately labeled and referenced in the text.

Experimental design

The study design is sound and methodologically appropriate for the research question. The authors have clarified previous ambiguities in their methodology and provided sufficient detail for reproducibility. Control groups, sample sizes, and statistical approaches are adequately described and justified.

Validity of the findings

The authors have strengthened the validity of their findings through additional analyses and clearer presentation of the results. The conclusions are well-supported by the data, and limitations are properly acknowledged. The revisions enhance the scientific rigor of the manuscript.

Additional comments

All revision requests have been satisfactorily addressed. I commend the authors for their thorough and thoughtful responses to the review comments. I find the revised version of the manuscript suitable for publication in its current form.